# Pepper Fruit Extracts Show Anti-Proliferative Activity against Tumor Cells Altering Their NADPH-Generating Dehydrogenase and Catalase Profiles

**DOI:** 10.3390/antiox12071461

**Published:** 2023-07-20

**Authors:** Marta Rodríguez-Ruiz, María C. Ramos, María J. Campos, Inmaculada Díaz-Sánchez, Bastien Cautain, Thomas A. Mackenzie, Francisca Vicente, Francisco J. Corpas, José M. Palma

**Affiliations:** 1Group of Antioxidants, Free Radicals and Nitric Oxide in Biotechnology, Food and Agriculture, Department of Stress, Development and Signaling in Plants, Estación Experimental del Zaidín (Spanish National Research Council, CSIC), 18008 Granada, Spain; marta.rodriguez@eez.csic.es (M.R.-R.); mariajesus.campos@eez.csic.es (M.J.C.); javier.corpas@eez.csic.es (F.J.C.); 2Department Screening & Target Validation, Fundación MEDINA, 18016 Granada, Spain; carmen.ramos@medinaandalucia.es (M.C.R.); thomas.mackenzie@medinaandalucia.es (T.A.M.); fvperez1958@gmail.com (F.V.); 3Evotec, University Paul Sabatier Toulouse III, 31100 Toulouse, France; cautainbastien@gmail.com

**Keywords:** antioxidants, capsaicin, catalase, hepatoma, hydrogen peroxide, NADPH, nitric oxide, pancreas, hot pepper

## Abstract

Cancer is considered one of the main causes of human death worldwide, being characterized by an alteration of the oxidative metabolism. Many natural compounds from plant origin with anti-tumor attributes have been described. Among them, capsaicin, which is the molecule responsible for the pungency in hot pepper fruits, has been reported to show antioxidant, anti-inflammatory, and analgesic activities, as well as anti-proliferative properties against cancer. Thus, in this work, the potential anti-proliferative activity of pepper (*Capsicum annuum* L.) fruits from diverse varieties with different capsaicin contents (California < Piquillo < Padrón < Alegría riojana) against several tumor cell lines (lung, melanoma, hepatoma, colon, breast, pancreas, and prostate) has been investigated. The results showed that the capsaicin content in pepper fruits did not correspond with their anti-proliferative activity against tumor cell lines. By contrast, the greatest activity was promoted by the pepper tissues which contained the lowest capsaicin amount. This indicates that other compounds different from capsaicin have this anti-tumor potentiality in pepper fruits. Based on this, green fruits from the Alegría riojana variety, which has negligible capsaicin levels, was used to study the effect on the oxidative and redox metabolism of tumor cell lines from liver (Hep-G2) and pancreas (MIA PaCa-2). Different parameters from both lines treated with crude pepper fruit extracts were determined including protein nitration and protein S-nitrosation (two post-translational modifications (PTMs) promoted by nitric oxide), the antioxidant capacity, as well as the activity of the antioxidant enzymes superoxide dismutase (SOD), catalase, and glutathione peroxidase (GPX), among others. In addition, the activity of the NADPH-generating enzymes glucose-6-phosphate dehydrogenase (G6PDH), 6-phosphogluconate dehydrogenase (6PGDH), and NADP-isocitrate dehydrogenase (NADP-ICDH) was followed. Our data revealed that the treatment of both cell lines with pepper fruit extracts altered their antioxidant capacity, enhanced their catalase activity, and considerably reduced the activity of the NADPH-generating enzymes. As a consequence, less H_2_O_2_ and NADPH seem to be available to cells, thus avoiding cell proliferation and possibly triggering cell death in both cell lines.

## 1. Introduction

Nowadays, cancer is considered one of the main causes of human death worldwide, as a consequence of the interaction of both environmental and genetic factors. Globally, cancer is a polyhedral pathology that implies both genetic alteration through mutations and uncontrolled cell proliferation [1]. It has been also shown that cancer is associated with the alteration of the nitro-oxidative metabolism in the cells and oxidative stress; oxidative damage episodes greatly contribute to the initiation, development, metastasis, and progression of cancer [1,2,3,4,5,6,7]. This situation of oxidative stress seems to down-modulate gene expression of mechanisms involved in DNA repair, cell proliferation, and the antioxidant machinery [1,6]. In fact, the analysis of the antioxidant enzymes’ profiles has been set as a promising prognostic index for the cancer of several organs like lung, bladder, ovaries, and colon [7,8]. Accordingly, strategies and therapies to promote an enhanced and appropriate antioxidant status in the cells have been considered as objectives to be exploited for the treatment of cancer. They include those based on the modulation of the cellular redox state [9,10], such as the epigenetic mechanisms driven by histone deacetylases [4], the photodynamic therapy to regulate the redox balance and the production of reactive oxygen species (ROS) [11], or those addressed to promote the ROS-associated simple summary autophagy, considering that the dysfunction in the autophagy is linked to cancer disease [12].

Alternatively, the supplementation of antioxidants or antioxidant-promoting compounds has been set. Thus, from in vitro and in vivo studies, it has been proposed that isoflavone genistein is a promising agent for cancer chemoprevention, with high potentialities for cancer therapy [13]. Likewise, the frequent use of vitamins C and E after the diagnosis of breast cancer has been associated with a lower probability of cancer recurrence, although the authors also showed that the effects of the antioxidants supplementation varied by the type of antioxidant [14]. In this sense, it has been also reported that a meta-analysis made from diverse databases did not support the correlation between antioxidants and the reduction of the risk of breast cancer [15]. Additionally, the lack of clear evidence for the protective role of selenium (Se) and antioxidants against gastrointestinal cancers has also been referenced [16]. Nevertheless, it was proven that dietary Se was oppositely associated with the risk of pancreatic cancer. Moreover, Se supplementation attenuated this association [17], thus confirming the important role of this element, necessary for the glutathione peroxidase function in order to remove H_2_O_2_. Thus, Saha and colleagues [18] found a correlation among nutrition, oxidative stress, and cancer initiation, and assessed the impact of the antioxidant capability against cancer therapy mediated by daily nutrition. In the same way, it was reported that the dietary intake of antioxidants from different plant foods was related to a reduction in the risk of gastric cancer [19].

The NADPH metabolism is also a key point in cancer episodes. Thus, for example, the over-expression of superoxide-generating NADPH oxidase (NOX) proteins in many tissues is associated with their own tissue and DNA damages promoted by ROS, which accompanies the pre-malignant conditions. It has been reported that several NOX isoforms produce ROS and contribute to the initiation and progression of a diversity of solid and hematopoietic malignancies, including colorectal and pancreatic carcinomas [20]. Conversely, it has been proposed that cancer cells require enhanced levels of NADPH for nucleotide synthesis and protection from ROS [21]. With these premises, a mechanism of NADPH homeostasis in cancer cells has been postulated [22], where enzymatic systems, such as those from the oxidative phase of pentose phosphate pathway glucose-6-phosphate dehydrogenase (G6PDH), 6-phosphogluconate dehydrogenase (6PGDH), as well as the NADP-isocitrate dehydrogenase (NADP-ICDH) and the NADP-malic enzyme (NADP-ME), could be involved [23,24,25,26,27,28,29].

From a human nutrition point of view, many plant species are sources of both macronutrients (carbohydrates, lipids, proteins, and fiber) and micronutrients with bioactive potential. Whereas macronutrients are mainly used for energy and structural purposes, micronutrients are essential in small amounts to modulate chemical reactions and metabolic processes. Within plant products, fruits and vegetables are useful vehicles for the provision of bioactive compounds, although in some cases, their specific actions and their identification are still undeciphered. In fact, plants contain a great number of compounds as part of their secondary metabolism, including vitamins (A, C, and E), carotenoids, polyphenols (flavonoids, anthocyanins, tannins, etc.), terpenoids, and alkaloids, among others. Most of these metabolites are abundant and well-recognized antioxidants in plant tissues, but their requirement in the human diet is very low since they mainly have regulatory roles [30].

Pepper (*Capsicum annuum* L.) is an herbaceous plant species which, aside from other relevant crops like tomato (*Lycopersicum esculentum*), potato (*Solanum tuberosum*), and aubergine (*Solanum melongena*), belongs to the Solanaceae family [31,32,33]. Pepper fruit is one of the most consumed horticultural products worldwide, with thousands of varieties cultivated nowadays throughout the world. Likewise, pepper fruit is characterized by its high amounts of antioxidants (vitamins C and A), carotenoids, polyphenols (including flavonoids), and capsaicinoids [34]. It has been reported that flavonoids, aside from antioxidant capacity, show antibacterial, antifungal, and anticancer effects [35,36,37,38,39,40]. Furthermore, capsaicin, an alkaloid that is exclusive of hot pepper fruits and responsible for their pungency, displays antioxidant, anti-inflammatory, antiproliferative, and analgesic activities. Recently, by the use of untargeted metabolomic approaches, it has been found that pepper (*Capsicum annuum*) fruits contain a series of compounds with potential therapeutic properties due to the presence, of quercetin and its derivatives, among others, with their content being modulated by nitric oxide (NO) [41]. In particular, capsaicin appears to play a role at the levels of the transient receptor potential vanilloid type-1 (TRPV1), and the phosphorylation of the tumor suppressor protein p53 [42,43,44,45,46,47,48]. Accordingly, due to the great and diverse consumption of pepper fruits worldwide and their molecular composition, these products might be considered to be vectors of bioactive compounds with therapeutic potential [31,49,50].

In this work, the anti-proliferative activity of crude extracts from four pepper fruit varieties (California, Padrón, Piquillo, and Alegría riojana) against seven tumor cell lines (lung, skin, liver, colon, breast, pancreas, and prostate) was investigated. The antioxidant status of liver and pancreas tumor cell lines, incubated with pepper fruit extracts from Alegría riojana, was then analyzed, and the activity profile of catalase, superoxide dismutase, glutathione peroxidase, and several NADPH-generating enzyme systems was determined. This allows an understanding of how the anti-proliferative activity exerted by crude extracts from pepper fruits is related to the antioxidant and redox metabolism of the susceptible tumor cells.

## 2. Materials and Methods

### 2.1. Plant Material and Preparation of Crude Pepper Fruit Extracts

Pericarp and placenta (Figure 1A) from pepper fruits of the California (Melchor), Padrón, Piquillo, and Alegría riojana varieties were used at both immature green and ripe red stages (Figure 1B). Plant material was powdered under liquid N_2_ using an IKA^®^ A 11 BASIC analytical mill (IKA Laboratories Inc.; Staufen, Germany). Basically, 1 g of powder was dissolved into 1 mL of pure acetonitrile (AcN), and the samples were shaken for 1 h at room temperature (RT) with a vortex at maximum speed, and then centrifuged for 45 min at 5410× *g* at RT. Afterward, samples were filtered and evaporated under a nitrogen gas stream. Finally, 200 µL of 20% (*v*/*v*) dimethylsulfoxide (DMSO) was added to each extract to prepare the pepper fruit homogenates.

### 2.2. Determination of Capsaicin

High-performance liquid chromatography-electrospray mass spectrometry (HPLC-ES/MS) was applied (HPLC-MS), using commercial capsaicin (Cayman Chemical, Ann Arbor, MI, USA) as the standard, as described earlier [51]. Three replicates from five pepper fruit samples (pericarp and placenta from the four varieties at the two ripening stages, green and red) were assayed. Basically, samples (0.5 g) were ground into a powder under liquid N_2_ and suspended into 2 mL AcN containing 100 ppm N-[(3,4-dimethoxyphenyl)methyl]-4-methyl-octanamide (DMBMO), as an internal standard. In our experimental conditions, an XBridge 2.1 × 10 mm pre-column and an XBridge 2.1 × 100 mm C18 3.5 µm column (Waters Corporation, Milford, MA, USA) were connected to an Alliance 2695 HPLC system coupled to a Micromass Quattro micro API triple quadrupole mass spectrometer (Waters). The retention time for capsaicin in our experimental conditions was 1.88 min, and the concentration of capsaicin was expressed as μg g^−1^ of fresh weight [51].

### 2.3. Tumor Cell Lines, Anti-Proliferative Activity Assays, and Preparation of Crude Extracts

The following cell lines were tested: Lung, A549; Melanoma (skin), A2058; Hepatoma (liver), Hep-G2; Colon, HT-29; Breast, MCF-7; Pancreas, MIA PaCa-2; and Prostate, PC-3. To determine the anti-proliferative activity, cells (about 10,000) from each line were incubated with several dilutions (1–1024 fold) of pepper fruit samples in a final volume of 100 μL in 96-well plates, and this experiment was performed three times. Different amounts (8–1000 μM/well) of commercial capsaicin (Cayman Chemical) were used as a positive control, considering previous reports on the anti-proliferative activity of this compound against tumor cell lines [42,43,44,45,46,47,48]. Then, the MTT [3-(4,5-dimethylthiazol-2-yl)-2,5-diphenyltetrazolium bromide] test was used after 72 h incubation of cells with pepper fruit extracts [52]. Under these conditions, the antiproliferative activity (0–100%) was determined. Also, in assays conducted for the analysis of the effect of pepper fruit samples (Alegría riojana variety) on the ROS and antioxidant metabolism of Hep-G2 and MIA PaCa-2 tumor cell lines, the IC50—as the amount of pepper fruit which reduced by 50% in the cell viability—was estimated in three independent experiments.

Regarding the biochemical assays after incubation for 3 d of tumor lines with pericarp of pepper fruits (0.12–0.15 g dry weight), cells (46 × 10^6^) from Hep-G2 and MIA PaCa-2 were resuspended and homogenized in an extraction buffer consisting of 50 mM Tris-HCl pH 7.8, 0.1 mM EDTA, 0.1% (*v*/*v*) Triton X-100, 10% (*v*/*v*) glycerol, and 1 mM PMSF. Then, crude extracts were centrifuged at 15,000× *g* for 30 min at 4 °C. Supernatants were used for further assays. The protein concentration in the supernatants was determined with the aid of a Pierce™ BCA Protein assay kit (Thermo Fisher Scientific, Inc., Waltham, MA, USA), which is based on bicinchoninic acid (BCA) for colorimetric detection and quantification of total protein, using bovine serum albumin as the standard [53].

### 2.4. Enzyme Activities

Catalase activity (EC 1.11.1.6) was determined by following the decomposition of H_2_O_2_ at 240 nm as described by Aebi [54]. NADP-dependent dehydrogenase (NADP-DH) activities were determined spectrophotometrically by recording the reduction of NADP at 340 nm for 30 min. Assays were performed at 25 °C in a reaction buffer (1 mL) containing 50 mM HEPES, pH 7.6, 2 mM MgCl_2_, and 0.8 mM NADP, and the reaction was initiated by the addition of the specific substrates. Thus, G6PDH (EC 1.1.1.49) activity was initiated with 5 mM glucose-6-phosphate; to determine 6PGDH (EC 1.1.1.44) activity, the substrate was 5 mM 6-phosphogluconate; and for NADPH-ICDH (EC 1.1.1.42) activity, the reaction was initiated by the addition of 10 mM 2R,3S-isocitrate [55,56].

For the analysis of the superoxide dismutase (SOD; EC 1.15.1.1) isoenzyme profile, 20 µg of protein from cell extracts were separated by non-denaturing PAGE on 10% acrylamide gels. The isoenzymes were detected using the staining method based on the photochemical reduction of nitroblue tetrazolium (NBT) by superoxide radicals (O_2_^•−^). Isoenzymes were visualized as achromatic bands over a purple background [57]. The nature of each SOD isozyme was determined through the incubation of gels with 5 mM KCN previous to the staining method. Thus, copper- and zinc-containing SODs (CuZnSODs) are inhibited by KCN, whereas manganese-containing SODs (MnSODs) are resistant to that inhibitor [51].

The analysis of the glutathione peroxidase (GPX; EC 1.11.1.9) isozymes was carried out as described by Lin et al. [58] with some modifications. Samples (30 µg of protein) were separated by non-denaturing PAGE on 6% acrylamide gels. After electrophoresis, gels were submerged for 20 min twice in 50 mM Tris-HCl, pH 7.9. Then, gels were soaked in the same buffer containing 13 mM GSH (reduced glutathione) and 0.004% (*v*/*v*) H_2_O_2_ with gentle shaking and darkness conditions for 10–20 min at room temperature. After rinsing with distilled water, gels were incubated with 1.2 mM NBT and 1.6 mM phenazine metasulfate (PMS) in distilled water, until the appearance of achromatic bands over a purple background. Finally, gels were washed with several changes of water for destaining [58]. 

### 2.5. SDS-PAGE and Immunoblot Analyses 

SDS-PAGE was performed in 4–20% precast polyacrylamide gels using a Mini-Protean IV electrophoresis cell (Bio-Rad, Hercules, CA, USA). For immunoblot analyses, proteins were transferred from gels onto PVDF membranes (0.45-µm) using a Trans-Blot^®^ Turbo^TM^ Transfer System (Bio-Rad). Cross-reactivity assays with different polyclonal and monoclonal antibodies were developed. Thus, nitrated proteins were detected by using a polyclonal antibody against nitrotyrosine (NO_2_-Tyr, Sigma Aldrich, Saint Louis, MO, USA) (dilution 1:500), and a monoclonal antibody against nitrotryptophan (NO_2_-Trp, Invitrogen, Thermo Fisher Scientific, Inc.) (dilution 1:1000); S-nitrosated proteins were detected by incubating with a polyclonal antibody against S-nitrosocysteine (SNO-Cys) (dilution 1:500); and for the detection of glutathionylated proteins, a polyclonal antibody against glutathione (anti-GSH) (dilution 1:1000) was used. The presence of MAPK/ERK kinases was detected using the monoclonal antibody anti-ERK1/2 (BDbiosciences, Franklin Lakes, NJ, USA) (dilution 1:1000).

For immunodetection, an enhanced chemiluminescence kit (Clarity™Western ECL Substrate, BioRad) was used according to the manufacturer’s instructions. Chemiluminescence was detected using a ChemicDoc^TM^ XRS imaging system (BioRad).

### 2.6. Antioxidant Capacity Assay

Total antioxidant capacity (TAC) was measured in cell samples (20 µL) using an e-BQC laboratory device (Bioquochem S.L., BQC Redox Technologies, Asturias, Spain) according to the manufacturer’s instructions. This method is based on the measurement of the redox potential sample, directly providing the antioxidant capacity as the charge (μC) of the electrons that are able to be donated to a free radical to be neutralized. A standard curve was made using ascorbate, and the values in each sample are expressed as ascorbate equivalents (AsA Eq, μM).

### 2.7. Statistical Analysis

One-way Anova and the Tukey test were used for the comparisons between means of capsaicin contents, using the Statgraphics Centurion program (Statgraphics Technologies, Inc., The Plains, VI, USA). For other parameters, the t-Student test was used. In both the Anova and t-Student, values for *p* < 0.05 were considered statistically significant.

## 3. Results

### 3.1. Crude Extracts from Pepper Fruit Show Anti-Proliferative Activity against Tumor Cell Lines

The anti-proliferative activity of crude extracts pericarp and placenta from green (immature) and red (ripe) pepper fruits of four varieties containing different capsaicin content (C, California; Pi, Piquillo; P, Padrón; AR, Alegría riojana) was assayed using seven tumor cell lines. The sequence of capsaicin content, as previously reported, was placenta from red AR fruits > placenta from red P > placenta from green AR = placenta from green P > pericarp from red AR > pericarp from red P > pericarp from green AR > pericarp from green P > placenta from green Pi > placenta from red Pi > pericarp from green Pi > pericarp from red Pi. None of the extracts from California, which belongs to the sweet pepper fruits, contained capsaicin (Figure 1C) [51]. Interestingly, it was demonstrated that green fruits from the four pepper varieties and with the lowest capsaicin levels displayed the highest anti-proliferative activity against the seven tumor cell lines, whereas samples with quite high capsaicin contents (above 250 μg/g fresh weight) barely reached 50% anti-proliferative activity (Figure 2). As a control, the assays were also performed using commercial capsaicin to test the negative effect of this compound in the viability of the used tumor cell lines. Thus, as given in Table 1, pure capsaicin displayed IC50 values ranging from 28.0–72.0 μM depending on the cell lines assayed. The amount of capsaicin loaded in each well is also indicated in the table, and it ranged from 0.86 to 2.2 μg.

In order to seek the specific mechanisms where this anti-proliferative activity resides, green pericarps from the cultivar Alegría riojana (with almost negligible capsaicin content) were set, and the IC50 index for Hep-G2 and MIA PaCa-2 tumor cells was determined. Thus, for Hep-G2, it was found that 15.48 mg of pepper fruit fresh weight provoked 50% lethality in 10,000 cells (Figure 3A), whereas in MIA PaCa-2 cells, this parameter was much lower, at 17.10 mg/10,000 cells (Figure 3B).

Then, the potential effect of crude pepper extracts on the tumor stage of cells was assayed by using an antibody against ERK1/2, two related protein-serine/threonine kinases which are used as markers of tumor events through their participation in the Ras-Raf-MEK-ERK signal transduction cascade [59]. Thus, as observed in Figure 4A, in Hep-G2 cells, this marker showed few changes, whilst in pancreas tumor cells, an enhancement of the protein with higher size was detected, as corroborated by the analysis of the intensity of bands through the program ImageJ (version 1.52p) (Figure 4B).

### 3.2. Pepper Fruit Extracts Slightly Alter the Nitro-Oxidative Status of Tumor Cells

Posttranslational modifications (PTMs) such as tyrosine (Tyr) and tryptophan (Trp) nitration, as well as oxidative PTMs (oxiPTMs), including S-nitrosation and S-glutathionylation, were analyzed to follow the nitro-oxidative status in tumor cell lines incubated with crude extracts from the Alegría riojana cultivar. Figure 5 and Figure 6 show the main polypeptides from both tumor cell lines which cross-react with antibodies against nitro-tyrosine (NO_2_-Tyr), nitro-tryptophan (NO_2_-Trp), nitroso-cysteine (CysNO), and glutathione. As observed, several proteins seem to be post-translationally modified by the treatment of cells with pepper fruit extracts. Thus, the expression of Tyr-nitrated proteins studied in Hep-G2 and MIA PaCa-2 tumor cell lines showed few differences in both cell lines after incubation with crude extracts from pepper fruits (Figure 5). As depicted in Figure 5A, in Hep-G2 cells, five major polypeptide bands (PHY 1–PHY 5; Appendix A) were detected, with PHY 4 being reduced about 35% after the treatment. Likewise, in MIA PaCa-2 cells, nine main polypeptide bands (PMY 1–PMY 9; Appendix A) were observed in both control and treated cells. In this latter case, PMY 1 showed about a 15% decrease after treatment, whilst a notable increase was observed in polypeptides PMY 6 (around 43%) and PMY 7 (about 44%) after cells were incubated with the fruit extracts.

The pattern of Trp-nitrated proteins from both cell lines was also investigated (Figure 5B). In the western blotting assays with Hep-G2 cells, eight polypeptide bands (PHW 1-PHW 8; Appendix A) were mostly visible after the use of the antibody against NO_2_-Trp, with no intensity changes after the incubation with pepper fruit extracts. Likewise, nine cross-reacting bands (PMW 1–PMW 9; Appendix A) appeared in MIA PaCa-2 control plants, and no significant changes took place in cells subjected to the treatment.

Concerning S-nitrosated proteins, different profiles were obtained depending on the cell lines (Figure 6A). Thus, in Hep-G2 cells, six polypeptides (PHC 1–PHC 6; Appendix A) cross-reacted with the antibody against nitroso-cysteine (SNO-Cys), with PHC 6 being the most abundant one. The incubation with crude pepper fruit extract did not provoke changes in the S-nitrosation pattern. In MIA PaCa-2 cells, five major polypeptides were visible (PMC 1–PMC 5; Appendix A), and no changes were detected either after the treatment. The potential PTM by glutathionylation was also followed in our experimental design. In Hep-G2 cells, nine polypeptides (PHG 1–PHG 9; Appendix A) were found to recognize the antibody against glutathione (Anti GSH), where PHG 1 was the most prominent. The number and intensity of these polypeptides did not change after the treatment of cells with pepper extracts (Figure 6B). Regarding pancreas cells, six polypeptides were found (PMG 1–PMG 6; Appendix A), and PMG 3 and PMG 6 notably decreased (about 36% and 55%, respectively) in cells subjected to pepper treatment.

### 3.3. ROS Metabolism in Tumor Cells under Pepper Fruit Treatment

Trying to unravel and understand how the oxidative metabolism was modulated under pepper treatment in both tumor cells lines, Hep-G2 and MIA PaCa-2, this work also focused on a set of antioxidant enzymes such as catalase (CAT), superoxide dismutase (SOD), and glutathione peroxidase (GPX). Additionally, the total antioxidant capacity (TAC) was evaluated in both lines. Thus, as shown in Figure 7A, this latter parameter showed an opposite trend after the treatment with pepper fruit extracts depending on the tumor line. Thus, in Hep-G2 cells, the treatment triggered a significant increase in TAC. Nevertheless, the opposite tendency was observed in MIA PaCa-2 cells, where the treatment caused a slight although significant decrease in this index.

Catalase activity, a peroxisomal enzyme associated with the scavenging of H_2_O_2_ that is converted into H_2_O and O_2_, was increased after the treatment with pepper fruits (Figure 7B). Thus, after the treatment of tumor cells, the activity rose by up to 54% in Hep-G2 and 182% in MIA PaCa-2 cells. Following further analysis of the H_2_O_2_ metabolism, the activity of the GPX isoenzymatic system under pepper fruit treatment was analyzed in both types of tumor cell lines by non-denaturing PAGE. Thus, four GPX isozymes were detected, but none of them showed noticeable changes after both Hep-G2 and MIA PaCa-2 cell lines were incubated with crude extracts from pepper fruits (results not shown). 

The SOD isoenzymatic pattern was also investigated as a consequence of the incubation of both cell lines with pepper fruit extracts. Overall, four SOD isozymes were identified: two manganese-containing SODs (MnSOD I and MnSOD II) and two copper zinc-containing SODs (CuZnSOD I and CuZnSOD II), according to their sensitivity to cyanide (Figure 8). MnSODs are resistant to CN^−^, whereas CuZnSODs are sensitive. In Hep-G2, MnSOD II and CuZnSOD II were only visible, whilst in MIA PaCa-2 cells, the four isozymes could be detected, with CuZnSOD I being the most abundant. None of these SOD isozymes were influenced by the treatment of cells with crude extracts from pepper fruits.

### 3.4. Pepper Fruit Treatment Disturbs the NADPH-Generating Systems in Tumor Cells 

The metabolism of NADPH has been reported to be of relevance in tumor cell lines. Thus, the behavior of the three main NADPH enzymatic sources was studied in this work. The activity of glucose-6-phosphate dehydrogenase (G6PDH), 6-phosphogluconate dehydrogenase (6PGDH), and NADP-dependent isocitrate dehydrogenase (NADP-ICDH) were significantly reduced after the incubation of both Hep-G2 and MIA PaCa-2 cell lines with crude extracts from Alegría riojana pepper fruits. The incubation with pepper fruit extract inhibited G6PDH activity up to 63% in Hep-G2 cell lines, and 40% in MIA PaCa-2 cell lines (Figure 9A). A similar tendency was observed in 6PGDH and NADP-ICDH; treatment of tumor cells with pepper fruit extract provoked significant inhibition of 6PGDH activity in both cell lines by up to ≈80% (Figure 9B). Likewise, Hep-G2 and MIA PaCa-2 subjected to treatment with pepper fruit extract showed significant inhibition of NADP-ICDH activity, by 77% and 48%, respectively (Figure 9C).

## 4. Discussion

Plants have always been considered as health sources for humans, due to their great amount of exclusive secondary metabolites, which positively interact with our physiological functions [30,60,61,62]. Capsaicin (8-methyl-N-vanillyl-trans-6-nonenamide) is an alkaloid compound with a phenyl-propanoid nature, which is exclusively present in fruits from hot *Capsicum* (pepper) species, and where the pungency trait from those products resides [51,63]. It has been shown that capsaicin shows antioxidant activity, but it also functions as an analgesic, with certain roles against infectious diseases in the nervous, cardiovascular, and immune systems, as well as in inflammation, obesity, and cancer episodes [41,43,44,64,65]. It has been reported that in many cancers, capsaicin displays proapoptotic activity promoted by the TRPV1 (transient receptor potential vanilloid type-1) channel. Furthermore, this chemical appears to also promote the phosphorylation of the tumor suppressor protein p53, which provokes its activation [41,43,45,46,47]. TRPV1 was identified as the neuronal receptor for harmful stimuli, which may allow the development of treatment of chronic pains including cancer [42,48]. Accordingly, in this study, the potential anti-proliferative activity of crude extracts from capsaicin-containing hot pepper fruits was evaluated using several tumor cell lines, and the influence of such pepper homogenates in the antioxidant metabolism of such pathogenic cells was investigated.

As a first unexpected discovery, it was found that the pepper fruits/tissues which contain the highest capsaicin levels [51] did not always display the greatest anti-proliferative activity. Conversely, the fruit portion that totally inhibited the growth of cell lines corresponded to the green pericarp from the four varieties used (California-Melchor, Piquillo, Padrón, and Alegría riojana). Capsaicin is synthesized in the placenta tissue of fruits, and it migrates towards the pericarp as the fruit ripens. Accordingly, in hot pepper fruits, the tissues show the following sequence from the most to the least concentrated capsaicin levels: placenta from ripe red fruits > pericarp from ripe red fruits > placenta from immature green fruits > pericarp from green fruits. Fruits from sweet pepper do not contain capsaicin in either the placenta or pericarp. Thus, in our study, the green pericarp from the sweet California-Melchor variety displayed the same high anti-proliferative activity as the green pericarp of Alegría riojana, a variety whose fruit is highly pungent when they ripen [51,63]. In our assay conditions, when the pericarp of green fruits from Alegría riojana where used to determine the IC50 index for Hep-G2 and MIA PaCa-2 cells, and considering the capsaicin content determined in such tissue previously (8.55 μg capsaicin/g tissue) [51], values of 0.13 and 0.15 μg per well, respectively, were estimated. When the assays were carried out with pericarps from red fruit of the same pepper variety (766.26 μg capsaicin/g tissue), about 9.58 μg capsaicin per well (100 μL) was used, a value much higher than that obtained when pure capsaicin was assayed (Table 1). However, very little anti-proliferative activity was shown by this last plant material. This indicates that, although capsaicin was clearly proven to display inhibitory effects on the viability of cells when this compound is present in the pepper fruit material, the effect does not parallel the capsaicin one; it rather seems that other compounds present in the green tissue are responsible for the anti-proliferative activity. Recently, a series of compounds with anti-carcinogenic potential or related to cancer episodes have been detected in pepper fruits through metabolomic approaches. It was reported that the concentration of those molecules was higher in immature green fruits than in ripe red ones, and they include quercetin, tryptophan (melatonin precursor), and phytosphingosine [41,63]. Thus, new perspectives in the search for plant metabolites with anti-tumor properties are opened through the combination of metabolomics and biological assays, a research field that can provide a panel of crops with nutraceutical potentialities.

Once preliminary and discriminating assays were performed, green fruits from the Alegría riojana variety were set to investigate their effect on the ROS and NADPH metabolism of tumor cells from the Hep-G2 and MIA PaCa-2 lines. Apparently, the progression of cancer in Hep-G2 cells does not appear to be affected by the treatment with the crude pepper extract, since no changes were observed in the ERK1/ERK2 levels. ERK1 and ERK2 are related protein-serine/threonine kinases that take part in the signal transduction cascade associated with a great number of processes, among which cell adhesion, cell cycle progression, cell migration, cell survival, differentiation, metabolism, proliferation, and transcription are the best known [59]. Furthermore, it has been reported that this transduction cascade is increased in about one-third of all human cancers [65]. In the case of MIA PaCa-2 cells, the higher abundance of the larger subunit may indicate that the tumor events are favored by the treatment, and more research is necessary to clearly define the real tumoral status of those cells.

Cancer processes are commonly linked to altered oxidative metabolism in the cells, with oxidative damages occurring through the initiation, development, metastasis, and progression of the pathology [1,2,3,4,5,6,7]. As a consequence of this oxidative stress, post-translational modifications (PTMs) promoted by nitric oxide (NO) have been reported to take place in cancer [66,67]. It has been also reported that the alteration in the profiles of the nitration, S-nitrosation, and carbonylation of proteins, as a consequence of a NO-dependent oxidative/nitrosative stress, was associated with a reduction of cell survival in a hepatoma cell line [68]. Nitration of proteins is an irreversible PTM that occurs by the addition of a NO_2_- group to either tyrosines (NO_2_-Tyr) or tryptophan (NO_2_-Trp) protein residues, whereas S-nitrosation occurs by the direct addition of NO to some cysteine residues in a reversible way [69]. The results reported show that, except for a few polypeptides, the profile of nitrated proteins at both tyrosine or tryptophan residues did not vary, and the same occurs in the case of S-nitrosated proteins. This indicates that, under our experimental conditions, no remarkable oxidative events seem to take place in either cell lines after the incubation with crude pepper fruit extracts. Likewise, no events of glutathionylation (another type of PTM characterized by the addition of glutathione to some cysteine residues) seem to occur after the treatment, although this PTM has also been related to cancer [70,71].

Since cancer causes alteration in the oxidative metabolism, a broader analysis of how the treatment with pepper fruit extracts influenced the oxidative and redox metabolism of the two cell lines was performed in this work. Thus, it was observed that the treatment provoked significant changes in the total antioxidant capacity of both cell lines, although with an opposite trend. We further studied the isozyme profiles of the superoxide dismutase and the glutathione peroxidase systems. In both types of cells, a different isoenzyme pattern was observed for SOD depending on the organ, either the liver or pancreas. SOD scavenges superoxide radicals (O_2_^•−^) and converts them into H_2_O_2_ plus O_2_; MnSOD II, and CuZnSOD II were shared by both organs, although the activity in MIA PaCa-2 was almost negligible. On the contrary, the most prominent SOD isozymes present in the pancreas cells were undetected in hepatoma cells. This is an interesting issue that deserves attention, since this may indicate different regulation at the organ level, and perhaps developmental level, and this could be essential for the progress of a localized cancer to metastasis. Thus, it has been reported that in colorectal adenocarcinoma and its liver metastases, the differences in mRNA and protein levels of SOD isoenzymes indicate that SOD takes part in the adaptation of tumor cells to oxidative stress. This situation helps keep a certain level of ROS, which is necessary for appropriate cell proliferation. Furthermore, the expression of SOD isoenzymes seemed to be regulated both at transcriptional and post-transcriptional levels [72]. The role of SODs, especially the MnSOD type, has been thoroughly reported as a therapy practice in cancer disease [73,74,75]. Regarding glutathione peroxidase, our results did not show visible differences in GPX isoenzymatic activity, either in Hep-G2 and MIA PaCa-2 cell lines. However, GPX activity has been reported to be modified in cancerous cells and tissues [76,77,78,79,80,81].

Catalase is the main H_2_O_2_ scavenging system, which in eukaryotes is localized in peroxisomes. Thus, by virtue of its enzymatic activity, catalase is called to be a key point in many processes where H_2_O_2_ may promote oxidative damages; also in those situations, this species plays a role as a signaling molecule in many physiological situations [82] and references therein. Catalase has been considered to be a remarkable enzyme system in the research of cancer and potential chemotherapy [73,83,84,85]. In our study, the treatment with pepper fruit extracts promoted a substantial increase of catalase activity in the two cell lines, which could scavenge the high H_2_O_2_ concentrations which lead to cancer-associated oxidative stress, and help to arrest the tumor’s evolution.

The NADPH metabolism has been proposed to play important roles in cancer [22], where NADPH-generating enzymes from the pentose phosphate pathway (G6PDG and 6PGDH), as well as the NADP-ICDH and NADP-ME, are relevant [23,24,25,26,27,28,29]. In this sense, the considerable decrease of the G6PDH, the 6PGDH, and the NADP-ICDH in both Hep-G2 and MIA PaCa-2 cells after the treatment with pepper fruit extracts implies lower NADPH availability in those cells, thus limiting their evolution to advanced tumor stages.

In Figure 10, a model of how Hep-G2 and MIA PaCa-2 tumor cell lines seem to operate after they are incubated with pepper fruit extracts is shown. The higher catalase activity contributes to diminishing the H_2_O_2_ levels. This, combined with lower G6PDH, 6PGDH, and NADP-ICDH activities which lead to a decreased NADPH generation, may trigger the stopping of tumor cell proliferation and further cell death.

## 5. Conclusions

Natural products are commonly considered health sources that can also contribute to preventing cancer and complementing targeted therapies. Capsaicin is a compound of alkaloid nature where the pungency trait of hot pepper fruits resides. This chemical has been proven to show antioxidant, anti-inflammatory and analgesic activities, as well as anti-proliferative properties against cancer. However, the results presented in this work show that crude extracts from hot pepper fruits with a high capsaicin content do not exhibit anti-proliferative activity against Hep-G2 and MIA PaCa-2 tumor cell lines. Moreover, those tissues which contain little or negligible capsaicin levels displayed the highest anti-proliferative activity. It indicates that other compounds different from capsaicin have this anti-tumor potential, and therefore, new research oriented towards the elucidation of the nature of those compounds should be promoted. Thus, a combination of (un)targeted metabolomics and biological assays can contribute to identifying those molecules from pepper fruits with anti-proliferative activity. Those compounds seem to be responsible for the increased catalase activity and lowered NADPH-generating enzyme activities, which seems to lead to lower hydrogen peroxide and NADPH levels, thus avoiding cell proliferation and triggering cell death in both cell lines. However, more investigation is necessary to decipher the intimate mechanism exerted by crude pepper fruit extracts. Additionally, the use of purified compounds, once they have been identified from pepper fruits, will allow us to envisage new cancer therapies, thus providing a panel of nutraceutical plant products.

## Figures and Tables

**Figure 1 antioxidants-12-01461-f001:**
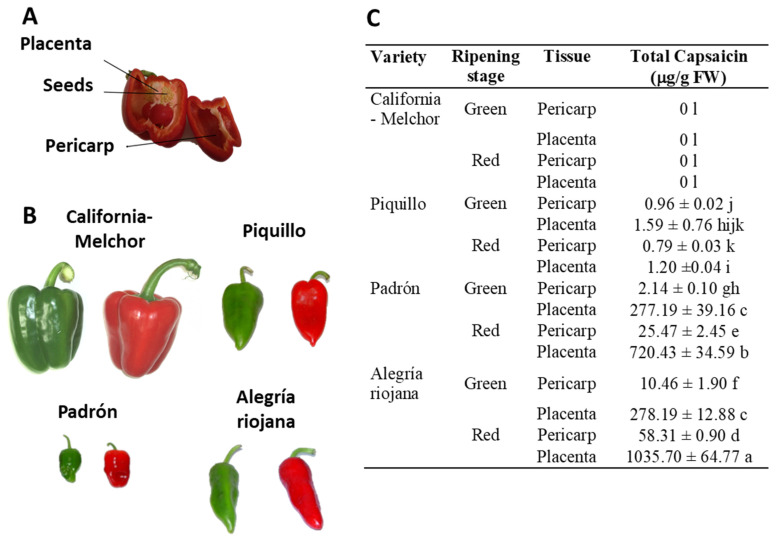
Plant materials and pepper varieties used in this work, and their capsaicin content. (**A**) Different parts of a representative pepper fruit. (**B**) Phenotype of fruits from the four varieties at two ripening stages: immature green and ripe red. Whereas California-Melchor is a sweet pepper fruit type, Piquillo Padrón and Alegría riojana contain different capsaicin levels with the sequence Piquillo <<< Padrón < Alegría riojana. (**C**) Content of total capsaicin levels in placenta and pericarp from fruits of the four pepper varieties at two ripening stages. Placenta tissue was processed once seeds were discarded. Data are the means ± SEM of three replicates determined from five fruits of the four varieties and at the two ripening stages. Different letters after each value indicate that differences were statistically significant (one-way Anova and Tukey test, *p* < 0.05). FW, fresh weight. Figure designed from data provided in Palma et al. [51].

**Figure 2 antioxidants-12-01461-f002:**
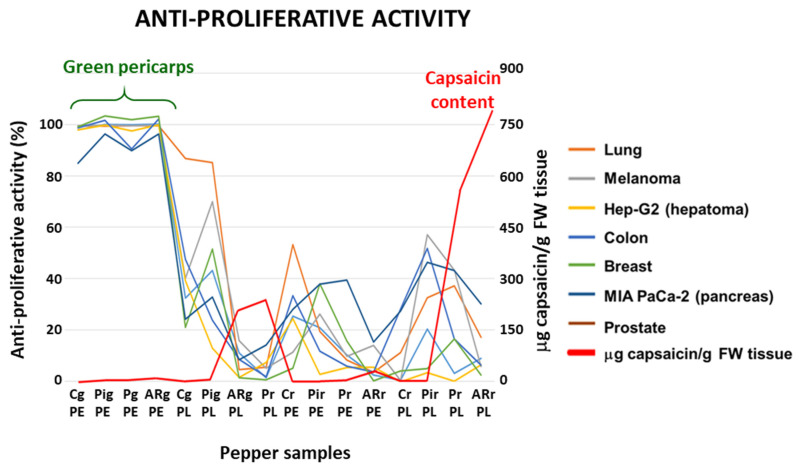
Antiproliferative activity and capsaicin content of crude extracts from fruits of four pepper varieties against seven tumor cell lines. The antiproliferative activity is expressed as % of dead cells after the treatment, and capsaicin content as μg/g fresh weight (FW), determined in the samples of pepper fruits used in the assays. Plant materials assayed in this experiment and their corresponding codes include pericarp (PE) and placenta (PL) from green (g) and red (r) pepper fruits from the varieties California (C), Piquillo (Pi), Padrón (P), and Alegría riojana (AR). The cell lines used corresponded to tumors from lung (A549), melanoma (A2058), hepatoma (Hep-G2), colon (HT-29), breast (MCF-7), pancreas (MIA PaCa-2), and prostate (PC-3). The plot shown is representative of three independent experiments. Color code of each graph is provided in the legend on the right.

**Figure 3 antioxidants-12-01461-f003:**
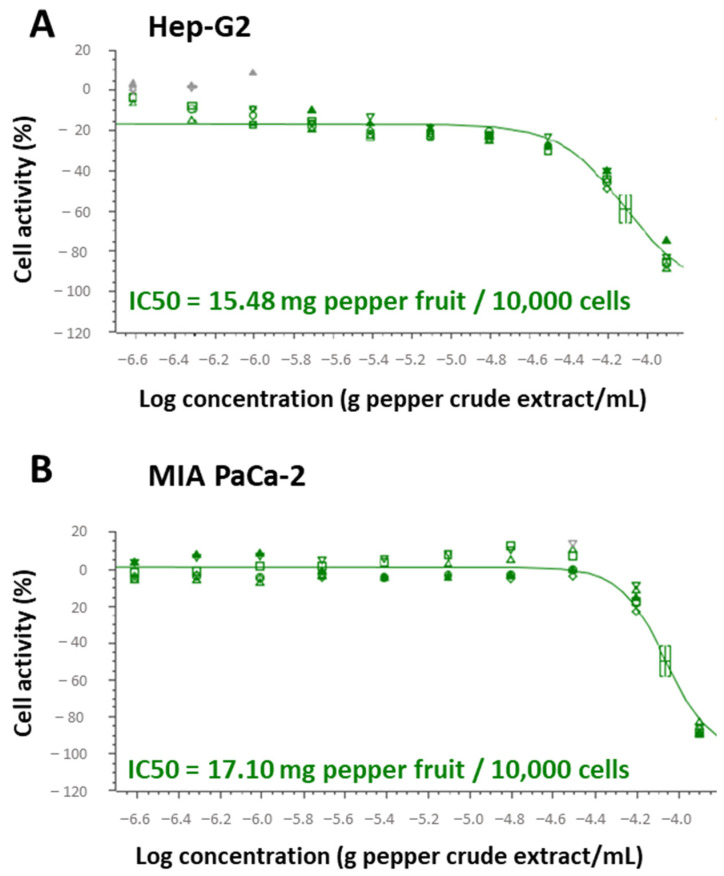
IC50 and level of the tumor marker ERK 1/2 from Hep-G2 and MIA PaCa-2 tumor cell lines after the incubation with crude pepper fruit extracts from the Alegría riojana variety. IC50 was defined as the amount of pepper fruit that reduced by 50% the cell viability. (**A**) IC50 in Hep-G2 cells. (**B**) IC50 in MIA PaCa-2 cells. The plots shown are representative of the three independent experiments performed. Each symbol at the assayed concentrations, either triangles, squares or circles, represents a replicate. Grey symbols denote outlier values once the regression curve is adjusted to all data. Square brackets in the two plots denote the IC50 for each cell line.

**Figure 4 antioxidants-12-01461-f004:**
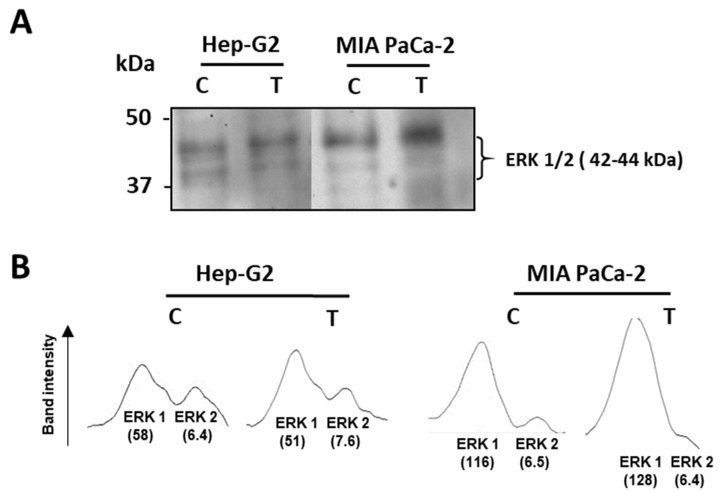
Level of the tumor marker ERK 1/2 from Hep-G2 and MIA PaCa-2 tumor cell lines after the incubation with crude pepper fruit extracts from the Alegría riojana variety. (**A**) Detection of the tumor marker ERK 1/2 (related protein-serine/threonine kinases 1 and 2) in Hep-G2 and MIA PaCa-2 cells that were untreated (C) and treated (T) with crude extracts from pepper fruits. Polypeptides were separated by SDS-PAGE in 4–20% precast polyacrylamide gels, and blotting assays were performed using a monoclonal antibody against ERK1/2 proteins (dilution 1:1000). Molecular weight markers are indicated on the left. (**B**) Densitograms of the ERK 1/2 bands detected after the blotting assay, where arbitrary units (in parentheses) were assigned to each tumor marker band by the use of the program ImageJ. The blotting image shown is representative of the three independent experiments performed.

**Figure 5 antioxidants-12-01461-f005:**
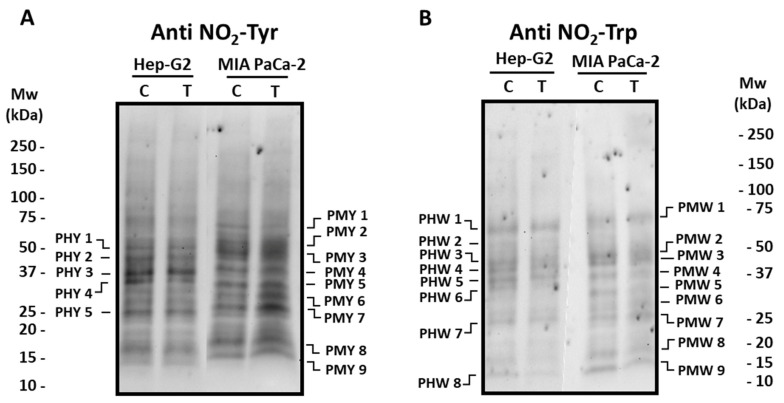
Detection of tyrosine (NO_2_-Tyr, panel (**A**)) and tryptophan (NO_2_-Trp, panel (**B**)) nitrated polypeptides from Hep-G2 and MIA PaCa-2 tumor cell lines after the incubation with crude pepper fruit extracts from the Alegría riojana variety. Polypeptides were separated by SDS-PAGE in 4–20% precast polyacrylamide gels, and blotting assays were performed using a polyclonal antibody against nitro-tyrosine (NO_2_-Tyr, dilution 1:500), and a monoclonal antibody against nitro-tryptophan (NO_2_-Trp, dilution 1:1000). C, untreated cells. T, treated cells. PHY, Polypeptide from Hep-G2 cells with nitrated Tyr (Y). PMY, Polypeptide from MIA PaCa-2 cells with nitrated Tyr (Y). PHW, Polypeptide from Hep-G2 cells with nitrated Trp (W). PMT, Polypeptide from MIA PaCa-2 cells with nitrated Trp (W). Molecular weight markers (Mw) are indicated on both sides of the panel. The blotting shown is representative of three independent experiments.

**Figure 6 antioxidants-12-01461-f006:**
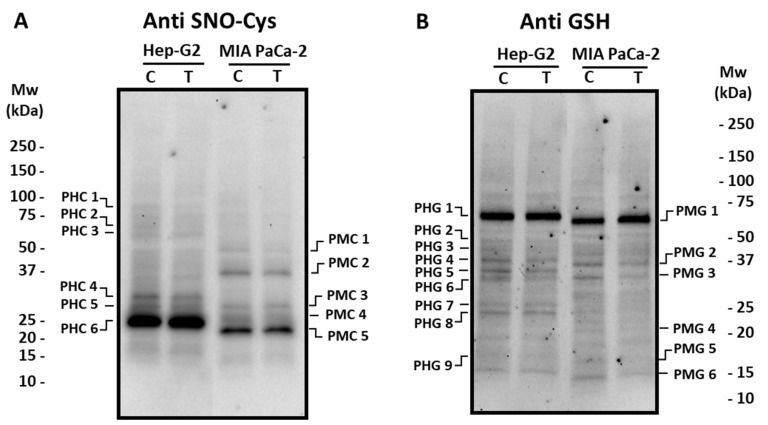
Detection of S-nitrosated (SNO-Cys, panel (**A**)) and glutathionylated (panel (**B**)) polypeptides from Hep-G2 and MIA PaCa-2 tumor cell lines after the incubation with crude pepper fruit extracts from the Alegría riojana variety. Polypeptides were separated by SDS-PAGE in 4–20% precast polyacrylamide gels and blotting assays were performed using a polyclonal antibody against S-nitrosocysteine (SNO-Cys, dilution 1:500), and a polyclonal antibody against glutathione (anti-GSH, dilution 1:1000). C, untreated cells. T, treated cells. PHC, Polypeptide from Hep-G2 cells with nitrosated Cys (C). PMC, Polypeptide from MIA PaCa-2 cells with nitrated nitrosated Cys (C). PHG, Polypeptide from Hep-G2 cells Glutathionylated. PMG, Polypeptide from MIA PaCa-2 cells Glutathionylated. Molecular weight markers (Mw) are indicated on both sides of the panel. The blotting shown is representative of the three independent experiments.

**Figure 7 antioxidants-12-01461-f007:**
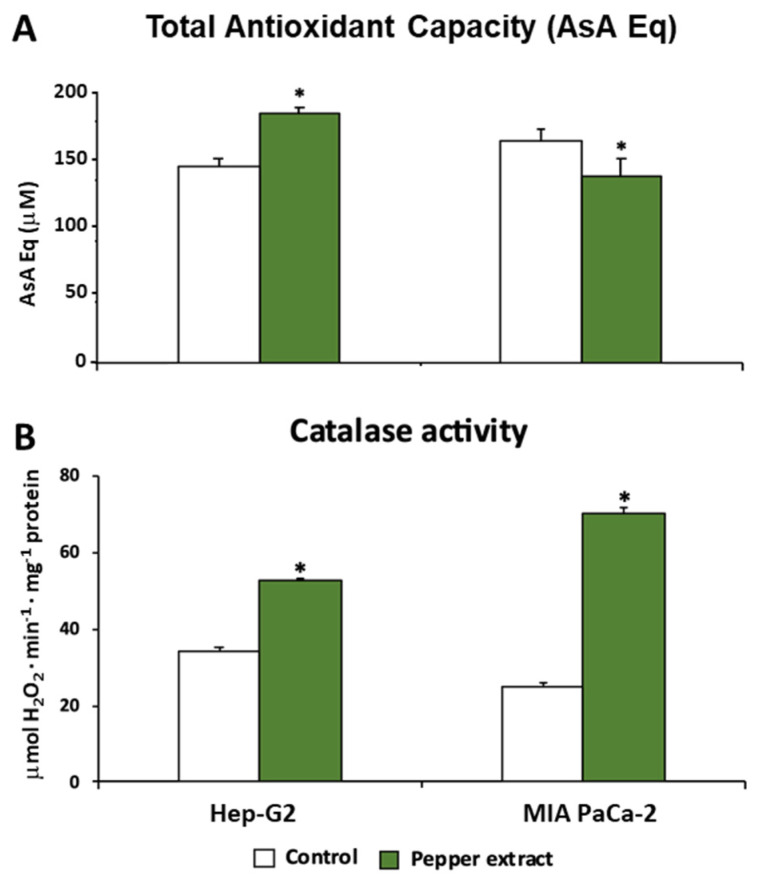
Total antioxidant activity and catalase activity from Hep-G2 and MIA PaCa-2 tumor cell lines after the incubation with crude pepper fruit extracts from the Alegría riojana variety. (**A**) Total antioxidant activity expressed as ascorbate equivalents. (**B**) Catalase activity. Data are expressed as the means of at least nine measurements from three independent experiments ± SEM. Asterisks denote significant differences in comparisons of treated cells to untreated (control) ones at *p* < 0.05 (Student *t*-test).

**Figure 8 antioxidants-12-01461-f008:**
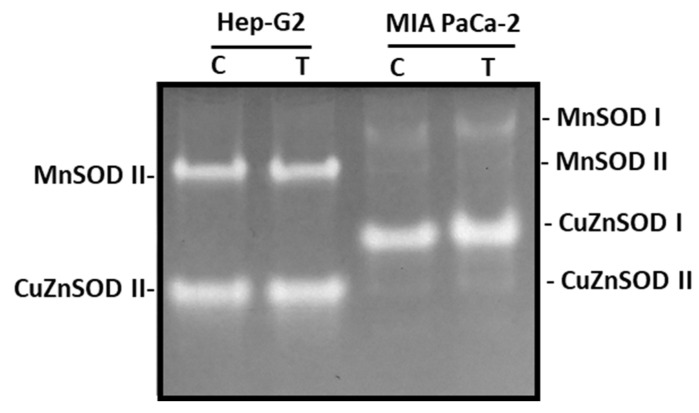
Superoxide dismutase (SOD) isoenzymes from Hep-G2 and MIA PaCa-2 tumor cell lines after the incubation with crude pepper fruit extracts from the Alegría riojana variety. Proteins (20 μg, see Section 2.4) were separated using non-denaturing PAGE in 10% polyacrylamide gels, and activity was detected in gels through the specific NBT staining method. C, untreated cells. T, cells treated with pepper fruit crude extracts. The zymogram shown is representative of the three experiments performed.

**Figure 9 antioxidants-12-01461-f009:**
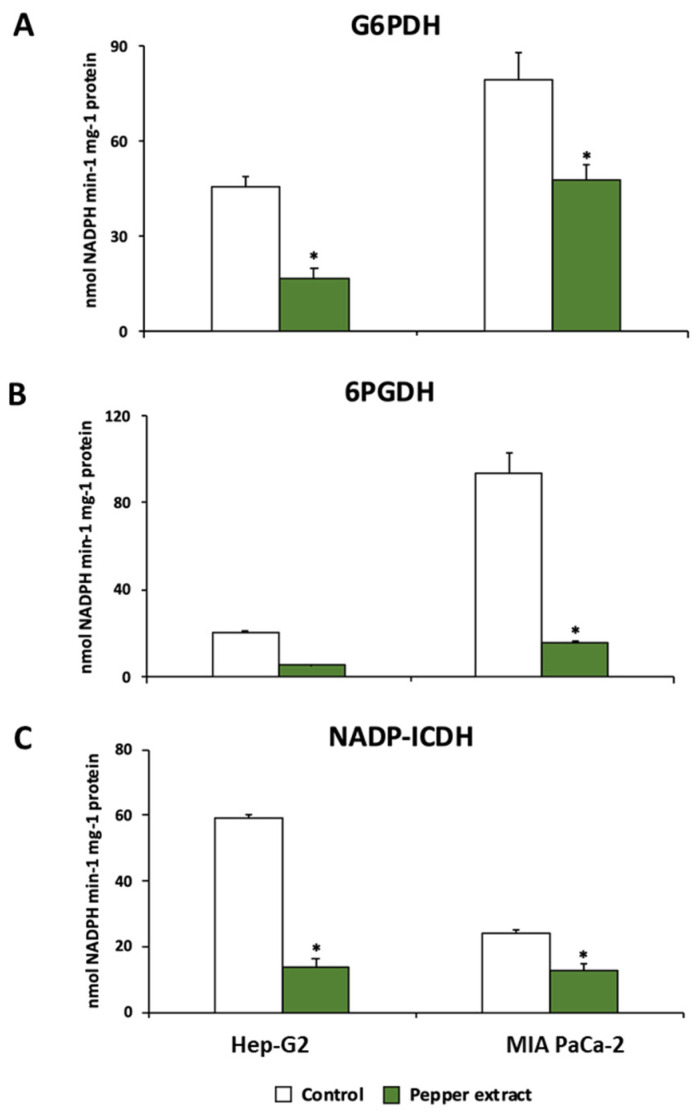
Activity of NADPH-generating enzymes from Hep-G2 and MIA PaCa-2 tumor cell lines after the incubation with crude pepper fruit extracts from the Alegría riojana variety. (**A**) Glucose-6-phosphate dehydrogenase (G6PDH). (**B**) 6-Phosphogluconate dehydrogenase (6PGDH). (**C**) NADP-dependent isocitrate dehydrogenase (NADP-ICDH). Data are expressed as the means of at least nine measurements from three independent experiments ± SEM. Asterisks denote significant differences in comparisons of treated cells to untreated (control) ones at a *p* < 0.05 (Student *t*-test).

**Figure 10 antioxidants-12-01461-f010:**
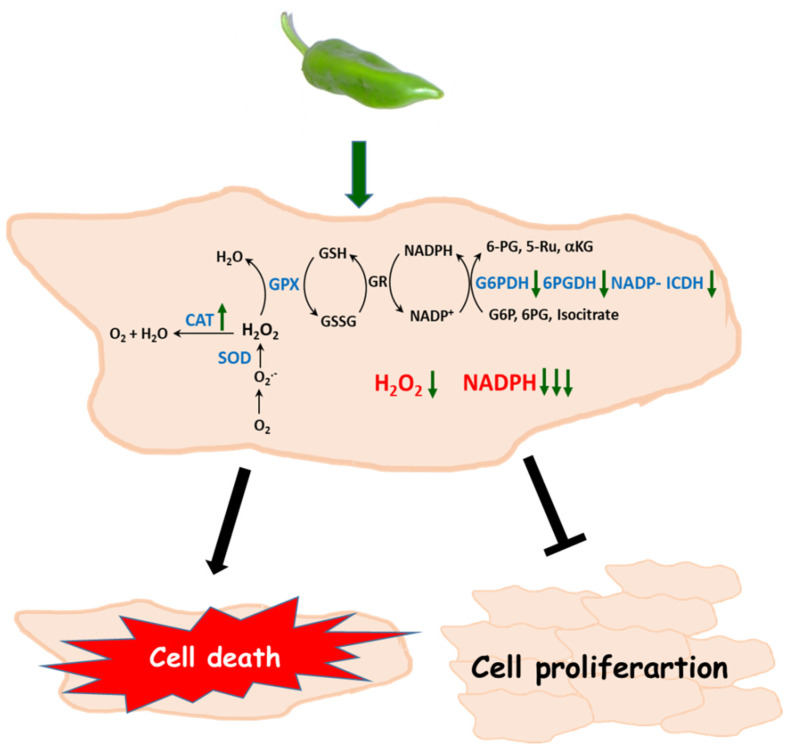
Proposed model of the effect of crude extracts from pepper fruits on the antioxidant and redox metabolism of tumor cells. As a consequence of the treatment with pepper extracts, an increase of catalase activity and a decline of the NADPH-generation enzymes occurs, thus provoking lower H_2_O_2_ levels and limited NADPH availability. Under these circumstances, cell proliferation is arrested and cell death may be triggered. CAT, catalase. SOD, superoxide dismutase. GPX, glutathione peroxidase. GSH, reduced glutathione. GSSG, oxidized glutathione. GR, glutathione reductase. G6PDH, glucose-6-phosphate dehydrogenase. 6PGDH, 6-phosphogluconate dehydrogensase. ICDH, isocitrate dehydrogenase. G6P, glucose-6-phosphate. 6PG, 6-phosphogluconate. Ru5P, ribulose-5-phosphate. αKG, α-ketoglutarate. In blue ink are the enzymatic systems studied in this work.

**Table 1 antioxidants-12-01461-t001:** IC50 determined for each of the tumor cell lines incubated with commercial capsaicin. IC50 was defined as the amount of capsaicin, prepared in methanol solution, which reduced the cell viability by 50%. Shown data are representative of three independent experiments.

Cell Lines	IC50 (μM Capsaicin)	Capsaicin in the Assay (μg/well)
A549	40.70	1.243
A2058	53.00	1.618
Hep-G2	28.00	0.855
HT-29	51.50	1.573
MCF-7	33.15	1.012
MIA PaCa-2	61.25	1.870
PC-3	72.20	2.199

## Data Availability

Not applicable. The data presented in this study are available in the own article, but also, if necessary, on request from the corresponding author.

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
