# Peer review of "Pepper Fruit Extracts Show Anti-Proliferative Activity against Tumor Cells Altering Their NADPH-Generating Dehydrogenase and Catalase Profiles"

_antioxidants, 2023, doi:10.3390/antiox12071461_

Round 1
Reviewer 1 Report
The paper, Pepper Fruit Extracts Show Anti-Proliferative Activity Against Tumor Cells Altering Their NADPH-generating Dehydrogenase and Catalase Profiles, makes an interesting point about the components of pepper fruits as anti-proliferative agents in tumor-derived cell lines, indicating that we do not know the active ingredients.
There are a few points in the data presentation that require improvement. Fig. 3c needs a graphic presentation of the levels.The tables presented in conjunction with Figs. 4 and 5 should contain the relative band intensity information.
I don’t know how to interpret Figs 7, top and bottom panels. The methods for the GPX panel have been around for many years with some updates in the methods paper you cite. Beutler and others used starch gels and overlays with GSH and tert-butyl hydroperoxide to detect alleles of GPX1 in the 1970s and map the gene (Beutler E, West C, Beutler B. Electrophoretic polymorphism of glutathione peroxidase. Ann Hum Genet. 1974 Oct;38(2):163-9. doi: 10.1111/j.1469-1809.1974.tb01947.x. PMID: 4467780.). So, there can be just GPX1 alleles in the mix. HepG2 also has high levels of GPX2 (maybe your GPX III), while MIAPaCa2 does not and neither express GPX3. It’s just unclear what I am to make of the gel. Also, we need to show standard dilution of both cell lines so we can see the sensitivity of the method and see how well calls of a difference, or no difference can be made. The SOD panel is cleaner in interpretation but also requires a reference dilution set.
The other data is cleaner and supports your major premise.
OK
Author Response
R1 Comments and Suggestions for Authors
The paper, Pepper Fruit Extracts Show Anti-Proliferative Activity Against Tumor Cells Altering Their NADPH-generating Dehydrogenase and Catalase Profiles, makes an interesting point about the components of pepper fruits as anti-proliferative agents in tumor-derived cell lines, indicating that we do not know the active ingredients.
Thanks a lot for this stimulating comment.
There are a few points in the data presentation that require improvement. Fig. 3c needs a graphic presentation of the levels.
Thank you. Figure 3 has been splitted into two figures, 3 and 4, to address this comment. So, regarding the former Fig. 3C, is now Fig. 4A, and it has been complemented with the image analysis where intensity of bands have been analyzed by ImageJ (new Fig. 4B), as indicated in the text (lines 307-308) and in the figure legend.
The tables presented in conjunction with Figs. 4 and 5 should contain the relative band intensity information.
Thank you. The relative intensity of differential bands has been provided based on the number of pixels in the supplementary tables, and appropriate comments have been added in that chapter. Please, be aware that these figures have also been complemented after adding the molecular weight markers
I don’t know how to interpret Figs 7, top and bottom panels. The methods for the GPX panel have been around for many years with some updates in the methods paper you cite. Beutler and others used starch gels and overlays with GSH and tert-butyl hydroperoxide to detect alleles of GPX1 in the 1970s and map the gene (Beutler E, West C, Beutler B. Electrophoretic polymorphism of glutathione peroxidase. Ann Hum Genet. 1974 Oct;38(2):163-9. doi:
10.1111/j.1469-1809.1974.tb01947.x. PMID: 4467780.). So, there can be just GPX1 alleles in the mix. HepG2 also has high levels of GPX2 (maybe your GPX III), while MIAPaCa2 does not and neither express GPX3. It’s just unclear what I am to make of the gel.
Thanks, this is an interesting comment. Starch gels have been used for long to investigate genetic polymorphysm of many enzymatic systems. This is a good method for that purpose, but very little resolutive since the pore size of gels is too big and isozymes are not discriminated quite well, since they are only separated by charge. For this same reason, this method is not appropiate either to quantify isozyme activity. On the contrary, polyacrylamide gels provide more functional data covering a higher number of isozymes, since proteins are separated based on their charge but also by their size and shape. It explains that Beutler and West only detected two bands in blood samples (red cells), whereas we were able to detect up to four isozymes in the analysis of different tissues: two in Hep-G2 cells (GPX III and GPX IV, liver), but three in MIA PaCa-2 (GPX I, GPX II, GPX IV, pancreas). The simplest comparison can be seen when talking of nucleic acids, which are much bigger than proteins. Usually, they are separated by agarose gels with high pore size. Neverthelss, when more reliability and accuracy are necessary in the case of some RNAs (that are smaller molecules than DNA) polyacrylamide electrophoresis is applied.
Also, we need to show standard dilution of both cell lines so we can see the sensitivity of the method and see how well calls of a difference, or no difference can be made.
To compare the isozyme activity among samples by polyacrylamide electrophoresis, aproximate similar protein amounts per sample must be loaded onto gels (30 mg, as indicated in the mansucript). It gives a comparison of the specific activity of each isozyme and, therefore if samples differ in this parameter. What Beutler and West did was to load the same GPX activity per sample. But, when the results in their gel are observed, we can see that not all samples show the same activity (intensity). So, there is no correlation between activity determined in the spectrophotometer and in gels. As indicated before, polyacrylamide gels are more convenient to detect funtional isozymes and to discriminate them better.
The SOD panel is cleaner in interpretation but also requires a reference dilution set.
See comment above.
The other data is cleaner and supports your major premise.
Thanks a lot for the valuable review.
Reviewer 2 Report
Rodríguez-Ruiz et al. present a work that exposed a variety of cancer cell lines to extracts obtained from different parts of a variety of peppers that were immature or ripe.
The cancer cell lines utilized included Lung, A549; Melanoma (skin), A2058; Hepatoma (liver), Hep-G2; Colon, HT-29; Breast, MCF-7; Pancreas, MIA PaCa-2; and Prostate, PC-3.
Pepper fruit pericarp and placenta of the varieties California (Melchor), Padrón, Piquillo, and Alegría riojana were used at both immature green and ripe red stages. The placenta tended to have a far greater concentration of capsaicin in all peppers with the exception of the Melchor, which had essentially no capsaicin.
The principle finding of this work was that capsaicin-independent inhibition of tumor cell proliferation was observed, with the extract of the pericarp being more effective than the placenta. Further, it was the least “hot” pepper that inhibited tumor growth the most. In particular, the authors focused on the inhibitor effects of hepatic and pancreas tumor cells. A variety of antioxidant effects and reduction in presumed nitrogen reactive species were posited for the overall decrease in tumor cell growth via a primary disturbance in NADPH-generating systems. It was odd that the total antioxidant capacity of liver tumor cells increased with pepper extract exposure but decreased in pancreatic cancer cells (figure 6). As summarized in diagrammatic form in figure 9, an increase of catalase activity and a decline of the NADPH-generation enzymes appears to halt cancer cell proliferation and perhaps accelerates cell death.
There are several issues with this work.
Methods
1. It is unclear how many experiments were performed for each series presented.
2. Statistics. The methods presented are brief, do not include an appropriate post hoc test for ANOVA, does not provide the rationale for the number of experiments based on statistical power, and does not include the number of experiments per condition per series performed.
Results
Figure 1. The ANOVA results within the table embedded in the figure are uninterpretable. There is no post hoc test to differentiate between the pepper results.
Figure 2. This figure is uninterpretable. The graphic text is too small to read on the X-axis, and there are too many relationships being compressed into this double-Y-axis presentation. Please split this up into multiple, more easily read, and interpreted figures.
Table 1. This is difficult to read secondary to the line numbers being interposed on the contents of the table.
Figures 4&5 are complicated and nearly uninterpretable. While it is possible to link the multiple labels of the gels with extracts, it is difficult.
Figure 6. While this figure mentions the number of experiments, n=3, is this sufficient to provide statistical power? Why only these few replicates? The same applies to figure 8.
Figure 7. How many times were these gels repeated? This question is also applicable to the other gels presented.
In summary, while the data presented may support the conclusions made, the methodology and data presentation weaken the ability of the authors to prove their case.
No comments beyond the review.
Round 2
Reviewer 1 Report
The general point of the paper, that the compounds modulating the anti-proliferative effects on cell are not capsaicin, is demonstrated. Some of the extra features are still a little had to understand. The obtuse explanation for interpreting figure 8 and the discussion makes me wonder why include it at all? I won’t claim to be an expert on GPX isoenzymes. But I was involved in much of the work that went into early characterization of GPX isoenzymes 1-4 in the late 1980s and early 1990s, both in cell lines (including HepG2) and animals. Your discussion of the figure makes no sense to me. We looked at HepG2 for isozyme expression and found both GPX1 and GPX2, able to separate them based on antibody pull down using anti-GPX1 and anti-GPX2 antibodies. Beutler’s use of red blood cells meant that he was only able to see GPX1 on his starch gels, so that variation in mobility was based on alleles and not isoenzyme forms. And I still object to the lack of any dilution series in both panels to help me understand how sensitive your assay is.
I recommend you remove the figure. The paper does not need that information. I think the paper should be published since the main point is otherwise supported.
English is just fine.
Paper is average for all points accept the items mentioned above, where it falls flat.
Author Response
Dear reviewer:
Thanks for your helpful comments. Following your suggestions, we remove the figure on GPX, but maintain the one on SOD, since to our knowledge, this is the first report on such system in MIA PaCa-2 cells and its comparison with that of Hep-G2. We hope the new version meets your query.
Best regards,
Pepe
Reviewer 2 Report
Thank you for addressing my concerns.
Author Response
Dear reviewer:
Thanks for your helpful comments to this manuscript.
Best regards,
Pepe
Round 3
Reviewer 1 Report
The authors addressed my concerns adequately and the paper should be accepted as is.
The authors addressed my concerns adequately and the paper should be accepted as is.